# Genome-Wide Analysis of AP2/ERF Gene Superfamily in Ramie (*Boehmeria nivea* L.) Revealed Their Synergistic Roles in Regulating Abiotic Stress Resistance and Ramet Development

**DOI:** 10.3390/ijms232315117

**Published:** 2022-12-01

**Authors:** Xiaojun Qiu, Haohan Zhao, Aminu Shehu Abubakar, Deyi Shao, Jikang Chen, Ping Chen, Chunming Yu, Xiaofei Wang, Kunmei Chen, Aiguo Zhu

**Affiliations:** 1Institute of Bast Fiber Crops, Chinese Academy of Agricultural Sciences, Changsha 410221, China; 2Department of Agronomy, Bayero University Kano, Kano PMB 3011, Nigeria; 3Key Laboratory of Genetic Breeding and Microbial Processing for Bast Fiber Product of Hunan Province, Changsha 410221, China; 4National Breeding Center for Bast Fiber Crops, Changsha 410221, China

**Keywords:** ethylene-responsive elements factor (ERF), ramie, stress response, ramet development, multi-omics analysis, waterlogging stress

## Abstract

AP2/ERF transcription factors (TFs) are one of the largest superfamilies in plants, and play vital roles in growth and response to biotic/abiotic stresses. Although the AP2/ERF family has been extensively characterized in many species, very little is known about this family in ramie (*Boehmeria nivea* L.). In this study, 138 AP2/ERF TFs were identified from the ramie genome and were grouped into five subfamilies, including the AP2 (19), RAV (5), Soloist (1), ERF (77), and DREB (36). Unique motifs were found in the DREB/ERF subfamily members, implying significance to the AP2/ERF TF functions in these evolutionary branches. Segmental duplication events were found to play predominant roles in the BnAP2/ERF TF family expansion. Light-, stress-, and phytohormone-responsive elements were identified in the promoter region of *BnAP2/ERF* genes, with abscisic acid response elements (ABRE), methyl jasmonate response elements, and the dehydration response element (DRE) being dominant. The integrated transcriptome and quantitative real-time PCR (qPCR) revealed 12 key *BnAP2/ERF* genes positively responding to waterlogging. Five of the genes are also involved in ramet development, with two (*BnERF-30* and *BnERF-32*) further showing multifunctional roles. The protein interaction prediction analysis further verified their crosstalk mechanism in coordinating waterlogging resistance and ramet development. Our study provides new insights into the presence of AP2/ERF TFs in ramie, and provides candidate AP2/ERF TFs for further studies on breeding varieties with coupling between water stress tolerance and high yield.

## 1. Introduction

Ramie (*Boehmeria nivea* L.) is a multipurpose crop, widely used as fiber, medicinal preparation, forage, phytoremediation, and as biofuel [1]. It has been cultivated in Asia for over 4000 years [2]. Ramie production is, however, ravaged by water stress [3], and the plant dies within 48 h of water submergence. An estimated 26.7% yield reduction was also reported after 7 days of drought stress [3]. Nowadays, the cultivation of ramie has transferred from good land to marginal lands, such as slope hillsides, poor lowlands, and heavy-metal-contaminated fields, leaving enough acreage for food crops in major producing countries. These land areas are constantly experiencing either water shortages, submergence, and nutrient deficiency, thus risking ramie production, which urgently calls for new resistant varieties that will withstand these threats. However, the genetic and molecular–physiological basis of stress tolerance of ramie is still largely unknown.

AP2/ERF is one of the most significant gene families in plants, encoding plant-specific transcription factors (TFs) that regulate signaling networks for various plant biological processes [4]. AP2/ERF TFs are mainly found in higher plants, and play a critical role in regulating environmental stress responses, including abiotic (temperature, salinity, and water stress) and biotic stresses [5,6,7,8]. Numerous reports showed that genetically modified plants of overexpressing AP2/ERF TFs exhibited increased tolerance to abiotic/biotic stresses [9,10]. AP2/ERF TFs are important regulators involved in plant growth and development, such as root initiation and development [11], stem growth [12], leaf size [13], control of flower growth and development [14,15], and fruit/seed development and maturation [16,17]. Recent studies have shown that AP2/ERF TFs are also involved in plant patterns of tillering or branch regulation [18,19]. These results suggest that AP2/ERF family genes should be unique gene resources for improving crop production coupling with stress tolerance, yet, little is known about this family in ramie.

In this study, we attempted to build an extensive map of the AP2/ERF gene family based on the third-generation sequencing data of ramie. We performed a genome-wide identification of AP2/ERF family genes, gene structure, chromosome distribution, gene duplication, *cis*-acting elements and transcription factor binding sites, the conserved motifs, and the phylogenetic relationships of their encoded proteins. In addition, expression patterns of the identified genes in different tissues and in response to water deficit, nitrogen deficit, and waterlogging stresses were analyzed. Key genes are identified and discussed in response to waterlogging stress and ramet development regulation.

## 2. Results

### 2.1. Identification, Characteristics, Phylogeny Relationship, and Classification of AP2/ERF Transcription Factors in Ramie

A total of 138 putative *AP2/ERF* superfamily TFs were identified from the ramie genome, and their sequence information and physical properties are shown in Appendix A. The number of amino acids varied from 109 to 819. The molecular weight and isoelectric point (pI) were 12,579.27–89,274 Da and 4.51–11.32, respectively. Most of the genes (112), as predicted, were localized in the nuclear region, with only 26 located extracellularly.

Phylogenetic analysis was performed with the 138 proteins of ramie and the 147 AP2/ERF superfamily TFs of *A. thaliana* using MEGAX software. The results show that the BnAP2/ERFs were clustered into four major clusters: Soloist, RAV (related to ABI3/VP), AP2 (APETALA2), and ERF (ethylene-responsive factors) subfamilies (Figure 1). The ERF cluster was further divided into ERF and DREB (dehydration-responsive element binding) subfamilies. The naming convention of AP2/ERFs of *Brachypodium distachyon* was followed for nomenclature [20]. The results of the analysis performed using the NCBI Batch CD-Search Tool confirmed the presence of the AP2 domain in each of the 138 identified proteins (Figure 2). According to gene structure and characteristics of the conserved domain sequences, thirteen proteins with two AP2 domains and six proteins (BnAP2-02, BnAP2-03, BnAP2-04, BnAP2-05, BnAP2-18, and BnAP2-19) with a single AP2 domain were classified as the AP2 subfamily. Five proteins with AP2 and B3 domains were classified as the RAV subfamily. BnSolo-01, which was highly homologous to At4g13040, was classified as the Soloist subfamily. Most identified proteins (113) were from the ERF family, with 36 and 77 DREB and ERF members, respectively. All of the DREB proteins had conserved amino acids at the 14th (V) and 19th (E) position, and several of these proteins showed E replaced by L, V, A, or Q (Appendix A). At the same time, the ERF proteins showed conserved amino acids at the 14th (A) and 19th (D) positions, with few having V and H in the places of A and D, respectively (Appendix A).

According to the classification of *A. thaliana* [21], the BnDREB subfamily was categorized into I–IV groups, and the BnERF subfamily into V–X groups (Table 1). None of the ramie members fell into the Xb-L group, as similarly reported in jute and rice [22].

### 2.2. Gene Structure and Conserved Motifs Analysis of BnAP2/ERFs

Gene structure analysis is crucial for understanding the structural diversity of genes, revealing such information as the position of the coding sequence (CDS) and the untranslated regions (UTR) (Figure 2). The number of introns in the 138 identified genes ranged from 0 to 12, of which 83 genes had no introns. Genes in the same phylogenetic group had a similar gene structure. For example, all members of the VII group (*BnERF-21*, *BnERF- BnERF-22*, *BnERF-23*, *BnERF-24*, and *BnERF-25*) had 5’ UTR, 3’UTR, and one intron, with the intron occurring in AP2 conserved domain, which suggests their critical function roles in ramie. Noticeably, all members of the AP2, RAV, and Soloist subfamilies in ramie had introns except *BnRAV-04*.

The putative conserved motifs (CMs), possibly reflecting the functional domains involved in specific protein binding sites, such as nucleases and transcription factors, are shown in Figure 3 and Appendix A. Members in the same groups generally had similar CMs, demonstrating structural similarities among the proteins. Some groups contained CMs specific to them, such as CMVII-4 and CMVIII-3, which were generally outside the conserved domain. The divergence of these CMs may be the cause of their functional differentiation.

### 2.3. Chromosome Distribution and Gene Duplication of BnAP2/ERFs

The identified BnA2/ERF superfamily TFs were unevenly located on the 14 chromosomes of ramie, except for *BnAP2-07*, *BnDREB-36*, and *BnERF-09* genes, which were not assembled (Figure 4). Eighteen *BnAP2s* were located on the chr6, while there were four genes located, each on the chr10 and chr11, respectively. Gene duplication events were also analyzed to understand the AP2/ERF gene family expansion and differentiation in ramie, and the results are shown in Figure 4. Eleven pairs of tandem duplicated genes were distributed on six chromosomes. Chr6 had four tandem duplicated pairs, chr1 had two, whereas the remaining (chr5, 6, 8, 12, and chr13) each had one. All tandem duplicated gene pairs fell into the III and IX subgroups. There were higher numbers of segmentally duplicated gene pairs (43) compared to the tandem, suggesting segmentally duplicated events as the main driving force in *BnAP2/ERFs* evolution.

Ka/Ks represents the ratio between non-synonymous substitutions (Ka) and synonymous substitutions (Ks), which can determine whether the gene encoding the protein is under selection pressure. Ka/Ks >> 1 stands for positive selection, Ka/Ks ≈ 1 represents neutral selection, and Ka/Ks << 1 means purifying selection [21,23]. With the exception of *BnERF-03/01* and *BnERF-05/01* duplicated pairs, which have a higher sequence divergence degree and farther evolutionary distance, the Ka/Ks ratio of all duplicated gene pairs was less than one (Appendix A).

The orthologous relationships between ramie and other plants, dicotyledons (*Arabidopsis thaliana* and *Cannabis sativa female*) and monocotyledons (*Oryza sativa* and *Zea mays*), were also investigated to further understand the evolutionary relationship of *AP2/ERF* genes (Figure 5). A total of 94 *BnAP2/ERF* genes had syntenic relationships with those of *A. thaliana* (65), *C. sativa* (74), *O. sativa* (33), and *Z. mays* (38)*,* respectively, among which 20 *BnAP2/ERF* genes were collinear with all four species (Appendix A). We can intuitively establish from Figure 5 that the orthologous gene pairs of ramie and dicotyledons were significantly higher than those of monocotyledons, and some collinear genes only existed between ramie and other dicotyledons.

### 2.4. The Putative Promoter Regions Analysis of BnAP2/ERF Subfamily

*Cis*-elements play critical roles in plant growth, stress response, and tissue-specific expression. A total of 55 *cis*-acting elements were retrieved from the promoter region of *BnAP2/ERF* genes, of which 28 were light-responsive elements, 11 were phytohormone-responsive elements, 9 were plant growth and developmental elements, and 7 were stress-responsive elements (Figure 6). Light-responsive elements were found in the promoters of all the *BnAP2/ERFs*, with the highest being G-box (466). Box 4 (423) was found in all the *BnAP2/ERFs* promoters, but that of *BnDREB-03*, *BnDREB-30*, *BnERF-06*, *BnERF-16*, *BnERF-73* and *BnAP2-01*. Among the phytohormone-responsive elements, BnAP2/ERFs contained abundant abscisic acid response elements (ABRE) (405) and methyl jasmonate response elements (CGTCA-motif and TGACG-motif) (398). Moreover, *BnAP2/ERFs* were found to contain many ARE elements for the stress-responsive elements (315), indicating that most BnAP2/ERF TFs may be induced by low oxygen stress, such as waterlogging. The significant presence of these elements suggests that most BnAP2/ERF TF members are involved in plant development and biotic/abiotic stress responses.

### 2.5. Gene Ontology Annotation and KEGG Enrichment Analysis of BnAP2/ERF Target Genes

Based on two approaches, 1700 genes with at least one DRE/CRT in their putative promoters and 1009 genes with at least one GCC-box in their promoters were manually searched from the ramie genome. After removing the redundant genes, 2197 potential *BnAP2/ERF* target genes were confirmed. It contained disease-resistance-related *PRs* genes, transcription factors such as *WRKY*, *NAC*, and *MYB*, as well as F-box proteins, *D14L*, *GLT1*, and *GRF1/3*, which may be related to tiller traits of plants.

The target genes were assigned to 54 functional groups and divided into 3 main ontologies: biological processes, molecular functions, and cellular components (Figure 7A). Some target genes were enriched in GO terms for immune system processes, response to stimulus, detoxification effects, antioxidant activity, and growth and development.

The most enriched KEGG pathways included riboflavin metabolism, cellular autophagy, gluconeogenesis, glutathione metabolism, and MAPK signaling pathways (Figure 7B). Many target genes were enriched in various metabolic pathways, transcription and repair, signal transduction, and environmental adaptation of KEGG pathways (Figure 7C). These results suggest that *BnAP2/ERF* genes are involved in biotic/abiotic stress responses as well as plant growth and development pathways.

### 2.6. Expression Analysis of BnAP2/ERF Genes Based on Transcriptome Data

#### 2.6.1. Tissue Specific Expression of BnAP2/ERF Genes

In this study, based on the expression profiles obtained from RNA-Seq analysis data, we constructed a heat map of the gene expression patterns in different tissues of ramie (Figure 8A). Thirty-nine genes were not expressed in any analyzed tissues, indicating that these genes may be pseudogenes or require certain specific developmental stages and environments for induction. The expression levels of different *BnAP2/ERFs* varied among tissues, with 70 genes expressed in all tissues (FPKM > 0). Seven genes (*BnDREB-06*, *BnDREB-08*, *BnERF-21*, *BnERF-30*, *BnERF-32*, *BnERF-51*, and *BnERF-77*) had high expression levels in all tissues, and most genes were expressed at higher levels in root and bast fiber than that in the stem and leaf.

#### 2.6.2. Expression Patterns of BnAP2/ERF Genes in Response to Various Abiotic Stresses

To determine the involvement of *BnAP2/ERF* genes in response to abiotic stresses, relative expression profiles in ramie leaves under water deficit, nitrogen deficit, and waterlogging stresses were constructed based on transcriptomic data (Figure 8B). Fifty-seven genes were differentially expressed (25 up-regulated and 32 down-regulated) under water deficit, most of which were members of I, II, V, and X groups. Among the down-regulated genes, the expressions of *BnDREB-11*, *BnERF-39*, and *BnERF-49* showed significant differences compared with those in the control (|log2FC| > 4). Seventy-two differentially expressed genes (DEGs) were detected under nitrogen deficiency stress, with 44 being up-regulated and 28 down-regulated. Members of the DREB subgroup showed a stronger response under nitrogen starvation than the other subgroups. Sixty-nine DEGs were detected under waterlogging stress, of which 49 members were up-regulated and 20 members were down-regulated. Some members of the DREB subgroup and most of the ERFs showed significant up-regulation, and most *BnAP2/ERF* genes showed positive feedback to waterlogging stress. Of particular interest is that the up-regulated genes were mainly from the ERF subfamily, and the down-regulated genes from the DREB subfamily.

Furthermore, *BnERF-17* and *BnERF-18* were up-regulated only under water stress (water deficit and waterlogging), while *BnERF-22*, *BnERF-23*, and *BnERF-25* in group VII were up-regulated only under waterlogging. *BnERF-30* and *BnERF-31* of the VIII group were up-regulated under waterlogging, and *BnERF-26* and *BnERF-32* showed up-regulation under both nitrogen deficit and waterlogging. Some genes up-regulated under waterlogging (*BnDREB-11*, *BnERF-02*, *BnERF-30*, *BnERF-32*, *BnERF-39*, *BnERF-40*, *BnERF-49*, *BnERF-50*, *BnAP2-02*) showed relative expression patterns in water deficit, further indicating that these genes have essential regulatory roles under water stress. On top of that, *BnERF-03*, *BnERF-07*, *BnERF-08*, *BnERF-10*, *BnERF-11*, *BnERF-18*, *BnERF-22*, *BnRAV-03*, *BnAP2-03*, *BnAP2-17*, and *BnAP2-19* were only expressed under abiotic stress, and no expression was detected under normal development, suggesting that these genes may play roles in response to abiotic stress.

#### 2.6.3. Expression Patterns of BnAP2/ERF Genes in Various Ramie Varieties with Significantly Different Ramet Numbers

Members in group VIII of the AP2/ERF superfamily, such as *AtESR1* (*AT1G12980*), *AtESR2* (*AT1G24590*), *AtPUCHI* (*AT5G18560*), *AtERF4* (*AT3G15210*), and *AtERF11* (*AT1G28370*) from *Arabidopsis*, have been reported to be involved in axillary meristem and branch development and stem elongation and growth [24]. Therefore, we hypothesize that the BnAP2/ERF members in group VIII may have a similar function in the regulation of ramet in ramie. The transcriptome data from two ramie varieties with different ramet numbers (Figure 9A) were used to evaluate the expression pattern of these *BnERFs.* The results (Figure 9B) show that *BnERF-26*, *BnERF-28*, *BnERF-30*, *BnERF-31*, *BnERF-32*, *BnERF-33*, *BnERF-34*, and *BnERF-35* were expressed in all tissues of the two varieties. *BnERF-27* was only minimally expressed in the root, while *BnERF-29*, *BnERF-36*, *BnERF-37*, and *BnERF-38* were not detected in any tissues. All detected genes were up-regulated in variety “Zhongzhu No.1” with a low ramet number, suggesting that they may be involved in the negative regulation of the ramet number. Based on these results, we speculate that these genes may have potential roles in the ramet traits of ramie.

#### 2.6.4. Verification of Gene Expression by qPCR

Considering that waterlogging can have a fatal impact on ramie in a very short time, and the enormous cost to be paid for establishing a new ramie field according to the perennial habit of the crop, expression profiles of *BnAP2/ERF* genes under waterlogging stress were analyzed by pot experiment in the present study. Based on the reported *Arabidopsis* homologs of members in group VIII in ramie, and the highly differentially expressed members in the transcriptome data of waterlogging stress, 12 *BnAP2/ERF* genes were identified as candidate genes involved in the waterlogging response.

Furthermore, qPCR analysis was performed to confirm the expression of the 12 candidate genes. The results (Figure 10) indicate that the expression of *BnERF-14*, *BnERF-21*, *BnERF-22*, *BnERF-24*, *BnERF-40*, and *BnERF-50* increased with the prolongation of stress duration under light. *BnERF-25* and *BnERF-32* showed an increasing and then decreasing trend in expression under stress induction, while *BnDREB-11*, *BnERF-39*, and *BnERF-49* decreased, only increased at 12 h. These results suggest that *BnAP2/ERF* members exhibited multiple roles under stress induction. Interestingly, instead of returning to normal expression in the recovery state after stress, some of the genes showed a significant increase. These genes may be related to plant growth and development after stress. The expression pattern of the genes under light avoidance conditions was significantly different from that under light. Some genes were not induced under darkness (*BnERF-21*, *BnERF-14*, *BnERF-39*, and *BnERF-40*), and some genes showed the exact opposite trend (*BnDREB-11*, *BnERF-22*, *BnERF-24*, *BnERF-25*, *BnERF-32*, *BnERF-40*, *BnERF-49*, and *BnERF-50*). *BnERF-26* showed strongly inhibited expression under stress as well as in the recovery state after stress.

The light-or-dark treatments significantly changed the patterns of gene expression profile under waterlogging. While some genes were not significantly induced by the waterlogging stress under darkness (*BnERF-21* and *BnERF-39*), some showed the exact opposite trend under both conditions (*BnDREB-11*, *BnERF-22*, *BnERF-24*, *BnERF-25*, *BnERF-32*, *BnERF-40*, *BnERF-49*, and *BnERF-50*). The expression of *BnERF-26* showed the same trend as that under the light condition. These results imply that *BnAP2/ERF* genes regulate ramie through a complex network under different conditions.

## 3. Discussion

### 3.1. Global Profile of AP2/ERF Gene Family of Ramie

The TFs are considered as ideal candidates for crop improvement because their overexpression enhances tolerances to multiple abiotic and biotic stresses in transgenic plants [25]. Extensive studies have demonstrated that AP2/ERF TFs are crosstalk factors in stress signal pathways involved in salicylic acid, jasmonic acid, ethylene, and abscisic acid signal transduction pathways [26], and hence play vital roles in regulating plant growth and development as well as in response to diverse stresses [27,28]. Although AP2/ERF TFs have been thoroughly identified and characterized in many plant species, very little is known about the *AP2/ERF* gene family in ramie. In the present study, we performed a comprehensive screening of *AP2/ERF* genes in the ramie genome, and performed analyses including genome-wide identification, gene structure, gene localization, *cis*-acting element analysis, motif analysis, *cis*-linkage, and downstream target gene function and expression patterns, which laid an important foundation for better understanding the molecular mechanisms of development and physiological adaptation in this crop.

The AP2/ERF superfamily genes are defined by the presence of at least one highly conserved AP2 DNA binding structural domain consisting of three *β*-folds and one *α*-helix of approximately 60 to 70 amino acids in length. Based on the number and sequence characteristics of the structural domains, the AP2/ERF superfamilies of ramie were divided into AP2 (APETALA2), RAV (related to ABI3/VP), DREB (dehydration-responsive element binding), ERF (ethylene-responsive factors) and Soloist subfamilies, which contain 18, 5, 36, 72, and 1 members, respectively. Our results also support the large difference in the number of DREB and ERF occurrences, while Soloist and RAV are small subfamilies with few members in land plants [29,30,31,32]. It is widely known that ancestral species have intron-rich genes, and most plant species have experienced extensive loss or insertion of introns due to selection pressure [33]. Our results show that almost all AP2 and Soloist subfamily genes of ramie, like in other plant species [20,32,34], had introns, whereas most DREB and ERF subfamily genes (71.3%) were intronless. Nevertheless, the RAV subfamily genes in ramie, unlike other plants, all have introns except *BnRAV-04*. The insertion of introns may be a special mechanism that developed during the evolution of the RAV subfamily in ramie to better adapt to survival pressures [35]. Most members of the AP2 subfamily contained two AP2 domains, while some had a single domain lacking a conserved WLG motif. The RAV subfamily contained two distinct DNA binding domains, AP2 and B3. The DREB and ERF subfamilies each contained two highly conserved amino acids located in the β-fold at the 14th position, valine/alanine (V14/A14), and at the 19th position, glutamate/aspartate (E19/D19), in the AP2 domain. These conserved amino acids may play an essential role in site-specific binding to DNA sequences, such as the dehydration response element (DRE)/C-repeat element (CRT) or the GCC-box in the promoter region of the target gene [36,37]. KEGG and GO analysis revealed that these target genes contained disease-resistance-related *PRs* genes, stress-related genes *RAB18*, *LEA3*, *TIP2*, and *POX2*, transcription factors such as *WRKY*, *NAC*, and *MYB*, as well as *F-box* genes, *D14L*, *GLT1*, and *GRF1/3*, which may be related to tiller traits of plants. *BnAP2/ERFs* may regulate ramie response to environmental pressures and ramet development by binding to the corresponding sites in the promoter regions of these genes.

An exciting feature of the AP2/ERF superfamily is that these members have transcriptional activation or repression activity. Amphipathic repressor (EAR) motifs found at the C-terminus of proteins are responsible for their transcriptional repression activity, which involves biotic and abiotic stress responses, plant internode elongation, and leaf senescence [38,39,40,41]. On the other hand, members lacking this motif had transcriptional activation activity. CMVII-4, a MCCGGAI(I/L) motif, was characteristic of the VII group members. Five ERF-VII members in *Arabidopsis* had N-terminal structures, and two of them (HRE1, HRE2) were anaerobic response proteins. In addition, numerous studies have shown that members in the VII group of the ERF subfamily play a central role in regulating waterlogging tolerance [42,43,44,45]. Therefore, we hypothesize that AP2/ERF TFs with these characteristics have the same functions in ramie, which provides useful information for the research of BnAP2/ERF genes.

In ramie, the number of members is similar to that in *Arabidopsis* and rice, suggesting that the number of AP2/ERF family members is relatively stable, independent of genome size, and that the difference in number may be due to expansion events during the evolution of different plant species. Nonetheless, gene evolution and duplication induced the differentiation of the numbers of *AP2/ERF* genes among plants. Thus, we calculated the Ka, Ks, and Ka/Ks ratios of *BnAP2/ERF* repeating gene pairs, including tandem repeats and fragment repeats, to estimate divergence time and selection pressure. All Ka/Ks values were below 1, suggesting that these genes might have experienced strong purifying selective pressure during evolution [46]. In addition, the gene density was 2.3734 *AP2/ERF* genes per Mb in ramie, while the values for rice and *Arabidopsis* were 0.4047 and 1.1760, respectively. These data indicate that the AP2/ERF family genes were retained during the evolution of ramie in the presence of extensive gene loss, which proves the significance of the AP2/ERF family in the growth and development of ramie [47,48].

### 3.2. The Roles of the BnAP2/ERF Gene Family in Responding to Abiotic Stresses

Plants must adapt to a variety of biotic/abiotic pressures because they are immobile throughout their life cycle. AP2/ERF genes play important roles in stress responses, including drought, waterlogging, heat, and salt stress. From the RNA-seq data, the DEGs of *BnAP2/ERFs* were identified. Under different pressures, each subfamily showed different behaviors. For example, under nitrogen deficit, the up-regulated genes were mainly in the DREB subfamily and the down-regulated genes were mainly in the ERF subfamily; this was the opposite for water deficit. However, the response genes under waterlogging were predominantly in the ERF subfamily. Furthermore, we found that *BnDREB-25*, *BnDREB-26*, *BnDREB-32*, *BnDREB-33*, and *BnDREB-34* in the DREB subfamily, *BnERF-04*, *BnERF-05*, *BnERF-09*, *BnERF-12*, *BnERF-13*, *BnERF-29*, *BnERF-36*, *BnERF-37*, *BnERF-38*, *BnERF-47*, *BnERF-53*, *BnERF-54*, *BnERF-56*, *BnERF-57*, *BnERF-71*, and *BnERF-72* in the ERF subfamily, *BnAP2-06*, *BnAP2-09*, *BnAP2-11*, *BnAP2-14*, *BnAP2-16*, and *BnAP2-18* in the AP2 subfamily, and *BnRAV-02* were not expressed under either normal developmental or abiotic stress conditions, suggesting that these genes may be pseudogenes. However, the possibility that these genes respond to other stressful environments cannot be excluded.

The involvement of *AP2/ERF* family genes in the waterlogging response has been reported for several crops, with the most attention being given to members of group VII. We found from qPCR results that group VII members such as *BnERF-21*, *BnERF-22*, *BnERF-24*, and *BnERF-25* were all up-regulated under waterlogging stress, with the latter three strongly induced even after reoxygenation, indicating their importance in stress tolerance and in growth and development. In addition, plants still have to balance the presence and absence of varying degrees of light under field conditions [49]. The results of *cis*-element analysis also show that the promoter region of the *BnAP2/ERFs* was enriched in light-responsive elements, in addition to elements such as ABRE and ARE, in response to biotic/abiotic stresses. This indicates that *BnAP2/ERFs* also play important roles in regulating plant development in response to photoperiod. Expression pattern analysis showed that some *AP2/ERF* genes had to be expressed earlier under shade conditions to cope with these stresses to ensure their growth and nutrient accumulation. It has been demonstrated that plants improve their tolerance to stress under different light environments by responding appropriately to abiotic stresses [50].

### 3.3. Candidate Genes for Improving Waterlogging Tolerance Coupling with Ramet Development

Plants actively slow down their growth when they are under stress, a complementary strategy they use to cope with adverse conditions [51]. While plants can adapt to adversity, this comes at the expense of yield under normal growth conditions. Therefore, under-standing the mutual synergy between the stress response and growth may be the key to resetting the stress–growth balance. Indeed, designing more stress-resistant but high-yielding crops is the target of breeders.

Ramie is a vital fiber crop that exhibits defeat, stalk shrinkage, and stunted chloroplast development in response to stress to ensure its growth [3,52]. Meanwhile, ramet has an important characteristic that is significantly related to the yield of ramie [53]. Our results show that the genes in group VIII of BnAP2/ERF played positive roles in coping with water stress and regulating the development of ramet. The homologs of *BnERF-27*, *BnERF-38*, *BnERF-36*, *BnERF-37*, *BnERF-32*, and *BnERF-30* were *AtESR1*, *AtESR2*, *AtPUCHI*, *AtERF4*, and *AtERF11* in *Arabidopsis,* respectively. However, the analysis of transcriptome data revealed that *BnERF-27*, *BnERF36*, *BnERF-37*, and *BnERF38* did not show differential expression in the two ramie varieties with different ramet numbers. Therefore, we predicted that *BnERF-30* and *BnERF-32* were likely to be involved in the multifunctional role of waterlogging tolerance and ramet development in ramie.

Protein interaction analysis is more intuitive and rapid for understanding gene function, and is also important for regulatory network relationships between functional proteins. We used STRING software to map an integrated protein interaction network based on AtERF11 and AtERF4. The results (Figure 11) show that they first formed a complex through their EAR-motif and TPL protein, which in turn bind to genes related to WUS [54] and the growth hormone pathway [55,56,57,58] to regulate tillering and branching in plants. These results reinforce the potential role of BnERF-30 and BnERF-32 in the ramet traits of ramie. In addition, the ERF-TPL complex can also interact with jasmonic-acid-pathway-related proteins to alleviate stress damage, such as JAZ1, NINJA, and TIFY8 [59,60,61].

Moreover, BnERF-32 (AtERF4) can directly bind to the repressor proteins SPL2 and SPL6 involved in axillary meristematic tissue to regulate plant structure. Studies have shown that the number of branches or tillers of some plants increases or decreases after waterlogging stress, which may be related to the growth regulation of the plants themselves after exposure to adversity. For example, waterlogging-tolerant species *Alternanthera philoxeroides* and *Hemarthria altissima* with well-developed root systems are more likely to grow new branches after waterlogging [62], while wheat grown in drylands exhibits a significant reduction in tillers [63]. Thus, *BnAP2/ERF* family genes may also act as a more complex crosstalk mechanism in coordinating stress and growth development.

## 4. Materials and Methods

### 4.1. Identification of BnAP2/ERF Gene Superfamily

The *AP2/ERF* gene sequences were identified from the ramie genome derived from the whole genome data sequenced by our laboratory. The hidden Markov model (HMM) file of the AP2 domain (PF00847) was obtained from the Pfam database (http://pfam.xfam.org/ (accessed on 18 April 2022)), and the target genes were searched using HMMER3.0 software with a threshold E-value < 0.05 [64]. To confirm that the gene sequences obtained were those of the target, the presence of the AP2 conserved domain was checked against the NCBI Batch CD-Search Tool with E-value < 0.01. The physical and chemical parameters of BnAP2/ERF proteins, including the number of amino acids, molecular weight, theoretical pI, instability index, aliphatic index, and grand average of hydropathicity (GRAVY) were predicted using the Expasy ProtParam (https://web.expasy.org/protparam/ (accessed on 28 April 2022)). The sub-cellular location was predicted using the Softberry ProtComp v.9.0 (http://www.softberry.com/ (accessed on 29 April 2022)).

### 4.2. Phylogeny, Conserved Motifs, and Gene Structure Analysis

*Arabidopsis thaliana AP2/ERF* protein sequences (147) were downloaded from TAIR (https://www.arabidopsis.org/ (accessed on 18 April 2022)) and used to establish the phylogeny. Multiple alignments of *BnAP2/ERF* and *AtAP2/ERF* genes were performed by MUSCLE with default parameters, and the result was pruned using the Trimmer function in TBtools [65]. A phylogenetic tree was constructed in the MEGA11 program using the neighbor-joining method with 1000 bootstrap and other default parameters [66,67]. The Interactive Tree of Life (iTOL) was used for visualization of the phylogenetic tree (https://itol.embl.de/ (accessed on 13 May 2022)) [68].

The MEME Suite 5.4.1 online tool (https://meme-suite.org/meme/tools/meme (accessed on 15 May 2022)) was used for predicting the BnDREB/ERF conserved motifs (CMs), with the maximum number of motifs set to 10 and other parameters set to default. The gene structure of *BnAP2/ERF* and the final overall figure was generated with Gene Structure View in TBtools [65].

### 4.3. Chromosome Distribution, Gene Duplication, and Evolutionary Analysis of AP2/ERF Homologous Genes

The ramie genome annotation files were used to obtain the *BnAP2/ERF* chromosome distribution. Gene replication within the ramie genome and between ramie and other species was analyzed using MCscanX software [69]. Advanced Circos and Multiple Synteny Plot functions of TBtools [65] were used for visualization. Using KaKs_Calculator 2.0 [70], we computed the synonymous (Ks) and non-synonymous (Ka) substitution of the *BnAP2/ERF* gene pairs.

### 4.4. Cis-Element Analysis

All sequences 2.0 kb upstream of the *BnAP2/ERF* genes were selected as promoter regions and submitted to PlantCARE (https://bioinformatics.psb.ugent.be/webtools/plantcare/html/ (accessed on 22 May 2022)) for prediction. The results are summarized in Appendix A.

### 4.5. KEGG Enrichment Analysis and Gene Ontology Functional Annotation of BnAP2/ERFs Target Genes

Target genes with DREB protein binding site elements DRE (TACCGACAT) and CRT (G/ACCGAC) were screened. In addition, genes with AP2/ERF TF binding site elements (GCC-box) with sequence “AGCCGCC” were manually searched and used in the subsequent enrichment analysis. Pathway enrichment analysis for the potential BnAP2/ERF target genes was conducted using the KEGG database [71]. Gene ontology (GO) annotations of the target genes were performed using the Blast2GO tool with default parameters [72]. The program’s output was divided into biological processes, molecular functions, and cellular components. The advanced 20 enriched KEGG pathways and GO enrichment analyses were plotted using the R package ggplot2.

### 4.6. Transcriptome Data Sources and Expression Analysis of the BnAP2/ERF Genes

Transcriptome analysis of tissue-specific expression was conducted using four different tissue samples of “Zhongzhu No.1” sequenced by our laboratory. Transcriptome data of water deficit and nitrogen deficit were obtained from [73]. The additional “Zhongzhu No.2” ramie transcriptome changes in leaves of 3-week-old seedlings exposed to waterlogging for 12 h and to normal conditions were profiled using RNA-Seq data in our recent research. In this study, transcript fragments per kilobase representing *BnAP2/ERF* expression levels per mapped read per million (FPKM) values were calculated from these transcriptome data. Heat maps were generated using TBtools with log_2_(FPKM + 1) normalization [65].

### 4.7. BnAP2/ERF Protein–Protein Interaction Network Prediction

The homology of *AP2/ERF* genes between *Arabidopsis* and ramie was determined using OrthoVeen2 [74]. The interaction predictions of BnAP2/ERF proteins with other proteins based on *Arabidopsis* homologs were obtained with high confidence of >0.400 using the STRING v11.5 online program, and used to build a correlation network [75]; the networks were visualized in Cytoscape v3.8.2.

### 4.8. Plant Materials and Sampling

The ramie seedlings of variety “Zhongzhu No.2”, which has the largest cultivation area in the world, were prepared by hydroponic methods. Seedlings with even sizes were transplanted in earthen pots in the greenhouse under controlled conditions of 60 ± 5% humidity at 30 ± 2 ℃, and cultured for more 3 weeks with only water. Then, the seedlings with similar height, leaf number, and leaf area were selected and divided into four groups with six biological replicates for each treatment. The CK group was grown by applying 100 mL nutrient solution every day and additional water to keep the soil moisture at 80% of soil field capacity for 10 days. The waterlogging group was grown under the same conditions as the CK group, but for 9 days, and the pots were immersed by maintaining 2 cm water level above the underlying soil surface [76] for 12 h, which was then drained. Samples of CK and waterlogging were collected at 0, 3, and 12 h after waterlogging treatment and 12 h after drainage. Two groups of identical waterlogging treatments were set up in light and dark environments, respectively [62]. The composition of the nutrient solutions of the treatments shown in Appendix A refer to Tan [77].

### 4.9. RNA Sample Extraction and qPCR Analysis

Samples from the waterlogging treatment were used for qPCR analysis. Total RNA was extracted using a SteadyPure Plant RNA Extraction Kit (Accurate Biotechnology (Changsha, China) Co., Ltd.). The RNA was reverse transcribed, using an Evo M-MLV One Step RT-PCR Kit (Accurate Biotechnology (Changsha, China) Co., Ltd.), into cDNA, and qPCR analysis was performed using gene-specific primers (Appendix A). The 18s gene was used as an internal control (Accession number: EU747115). The qPCR was performed using an SYBR^®^ Green Premix Pro Taq HS qPCR Kit II (Accurate Biotechnology (Changsha, China) Co., Ltd.) on a CFX96 Touch Deep Well Real-Time Quantitative PCR System (Bio-Rad) following standard procedures. Relative transcript levels were calculated using the 2^-ΔΔCt^ formula, and the results were depicted in histograms drawn using GraphPad Prism v8.0 software.

## 5. Conclusions

A comprehensive analysis of the AP2/ERF family genes in ramie was carried out in this study. A total of 138 *BnAP2/ERF* genes were identified using bioinformatics, and further classified into five subfamilies based on conserved sequence characteristics, gene structure, and motif constituent. Synteny analysis revealed that fragment replication events were the key drivers of *BnAP2/ERF* evolution. Analysis of gene promoter cis-acting elements and target gene predictions indicated that BnAP2/ERF members actively respond to plant growth and stress stimuli such as water stress, mineral substance stress, and temperature stress. Based on the expression patterns analysis of *BnAP2/ERFs*, we identified some key genes involved in water stress and ramet development. In addition, transcriptome data and results from the protein interaction network analysis suggest that *BnERF-30* and *BnERF-32* had multifunctional regulatory effects on waterlogging stress and ramet development in ramie. In short, these results guide us a step further towards understanding the basic information of the *AP2/ERF* genes in ramie, which might serve as a first step toward the comprehensive functional characterization of the *AP2/ERF* gene family via genetic approaches in the future.

## Figures and Tables

**Figure 1 ijms-23-15117-f001:**
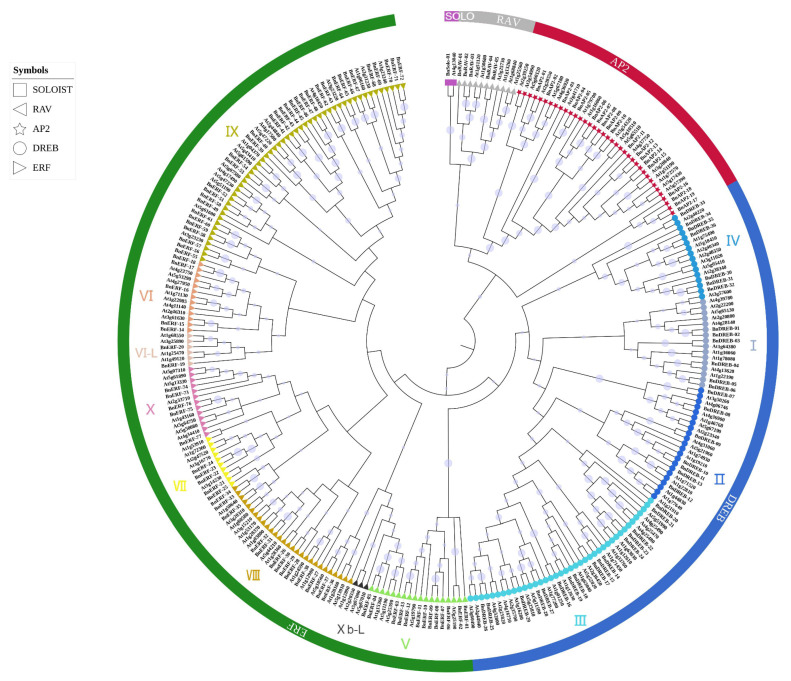
An unrooted phylogenetic tree constructed by the neighbor-joining method using AP2/ERF transcription factor domains in ramie and *Arabidopsis*.

**Figure 2 ijms-23-15117-f002:**
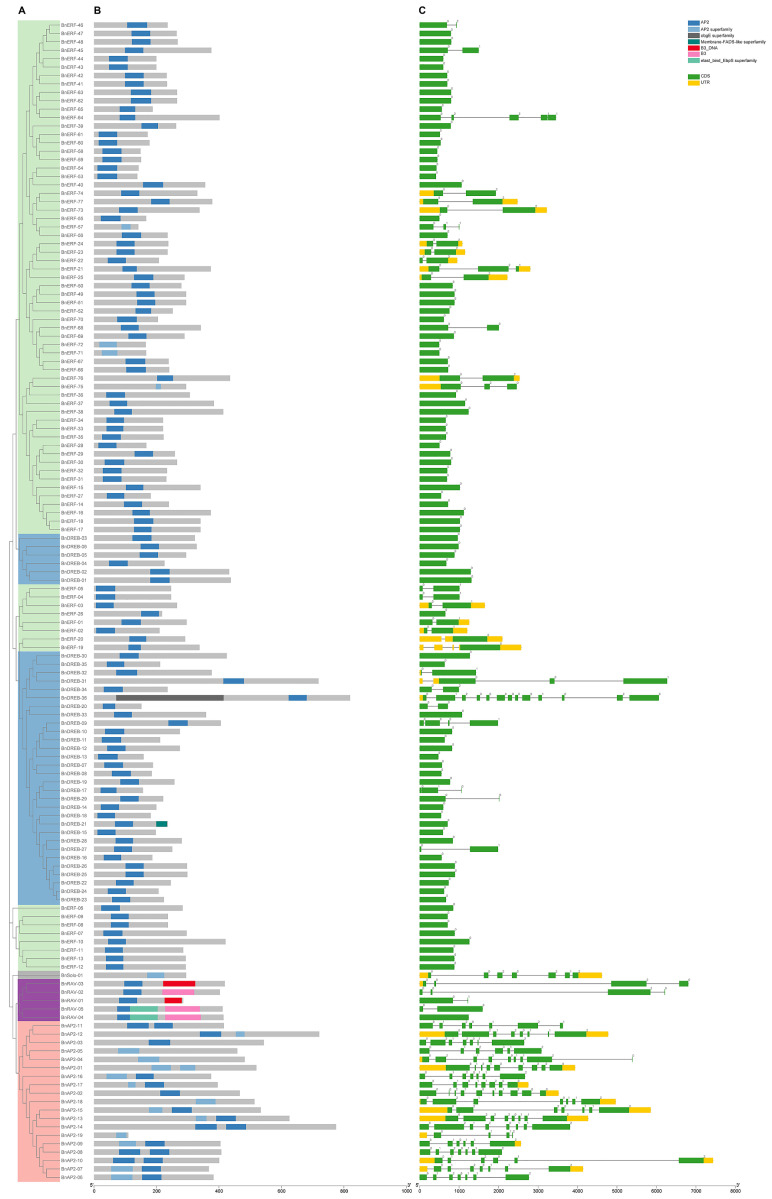
The phylogenetic tree, conserved domain, and gene structure of the AP2/ERF superfamily in ramie: (**A**) The neighbor-joining tree with 1000 bootstrap replicates of all BnAP2/ERF proteins. (**B**) The conserved domain identified by the NCBI Batch CD-Search Tool. The blue blocks are the AP2 domain, red blocks are the B3 domain. (**C**) The gene structure of *BnAP2/ERF* genes. The green blocks are CDS, yellow blocks are UTR, and grey lines are introns. The number stands for the phase information of intron–exon junctions.

**Figure 3 ijms-23-15117-f003:**
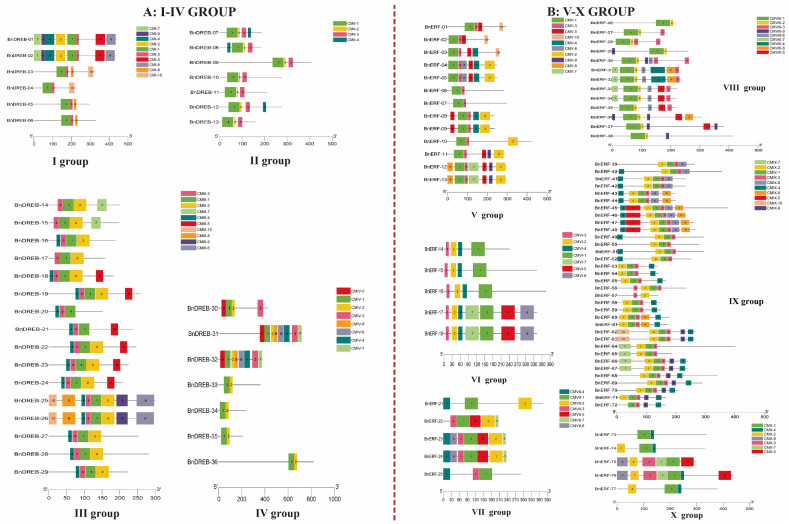
The composition of conserved motifs in ERF subfamily proteins of ramie: (**A**) I–IV group for the BnDREB subfamily; (**B**) V–X group for the BnERF subfamily.

**Figure 4 ijms-23-15117-f004:**
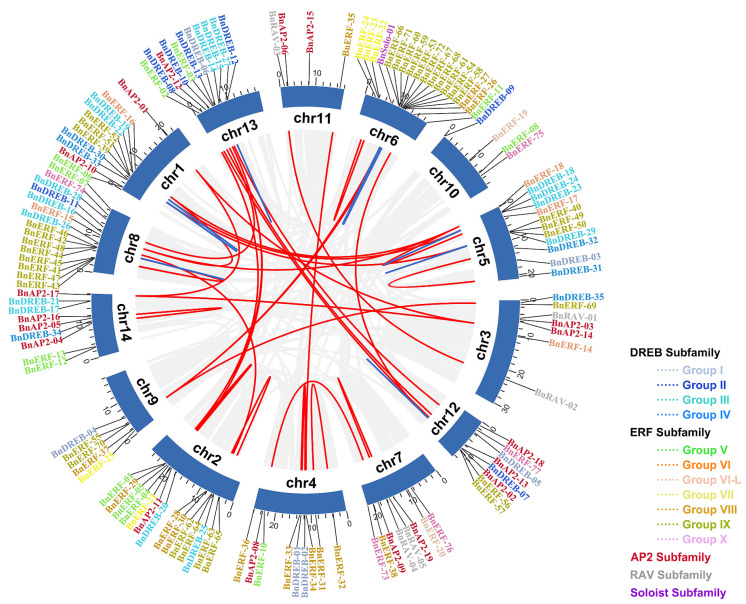
Distribution and gene duplication of BnAP2/ERF genes on 14 ramie chromosomes. Blue lines represent tandem duplicates of BnAP2/ERF gene pairs, red lines represent segmental duplicates of BnAP2/ERF gene pairs, and grey lines represent all synteny gene pairs in the ramie genome.

**Figure 5 ijms-23-15117-f005:**
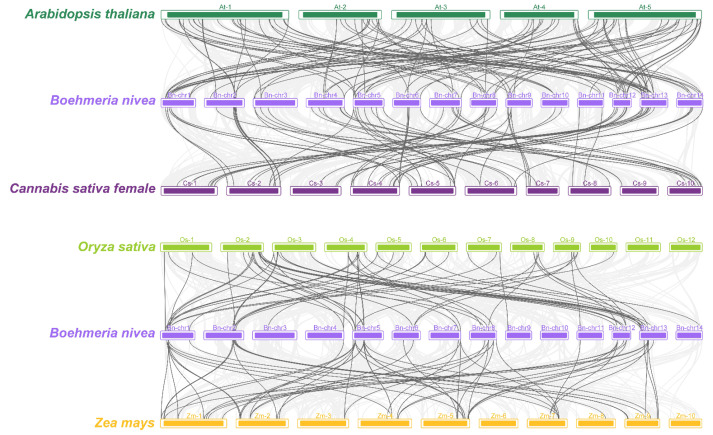
Gene duplication and synteny relationship of *BnAP2/ERF* genes between ramie and *A. thaliana*, *C. sativa*, *O. sativa*, and *Z. mays*. The deep gray lines stand for the syntenic *AP2/ERF* gene pairs, and the light gray stands for collinear blocks.

**Figure 6 ijms-23-15117-f006:**
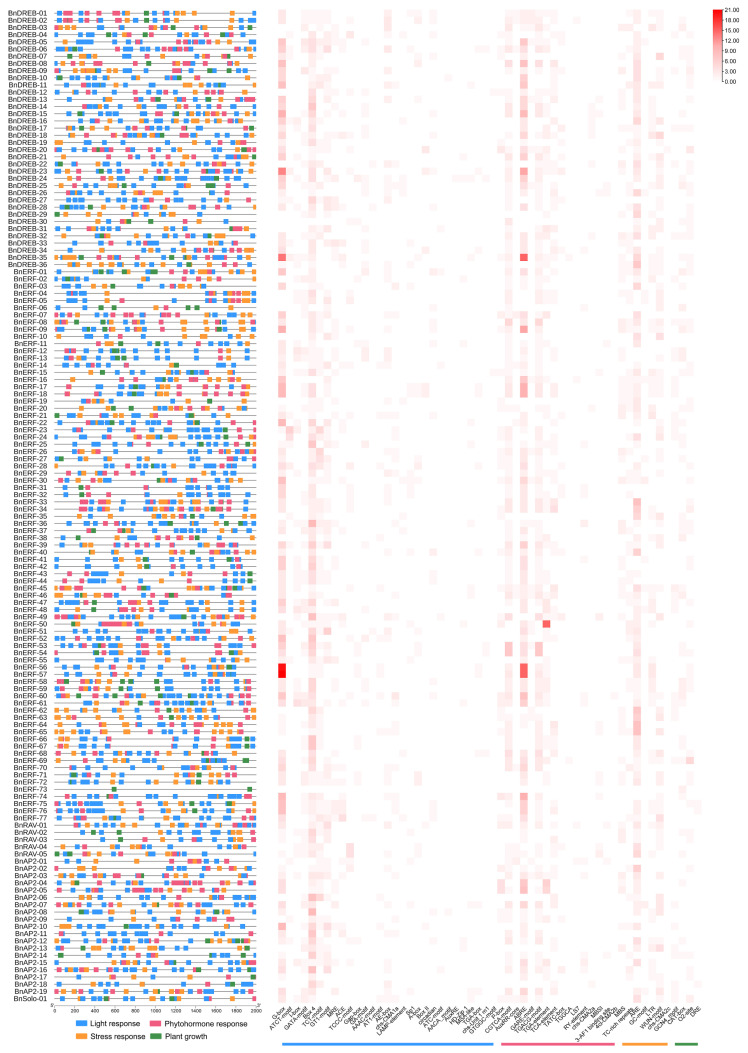
Analysis of *cis*-acting elements in the promoter region of *BnAP2/ERFs*. The left panel shows the distribution of *cis*-acting elements in the promoter region. The heat maps of cis-acting elements for the light-responsive, phytohormone-responsive, stress-responsive, and the color concentration of the squares indicates the number of *cis*-acting elements.

**Figure 7 ijms-23-15117-f007:**
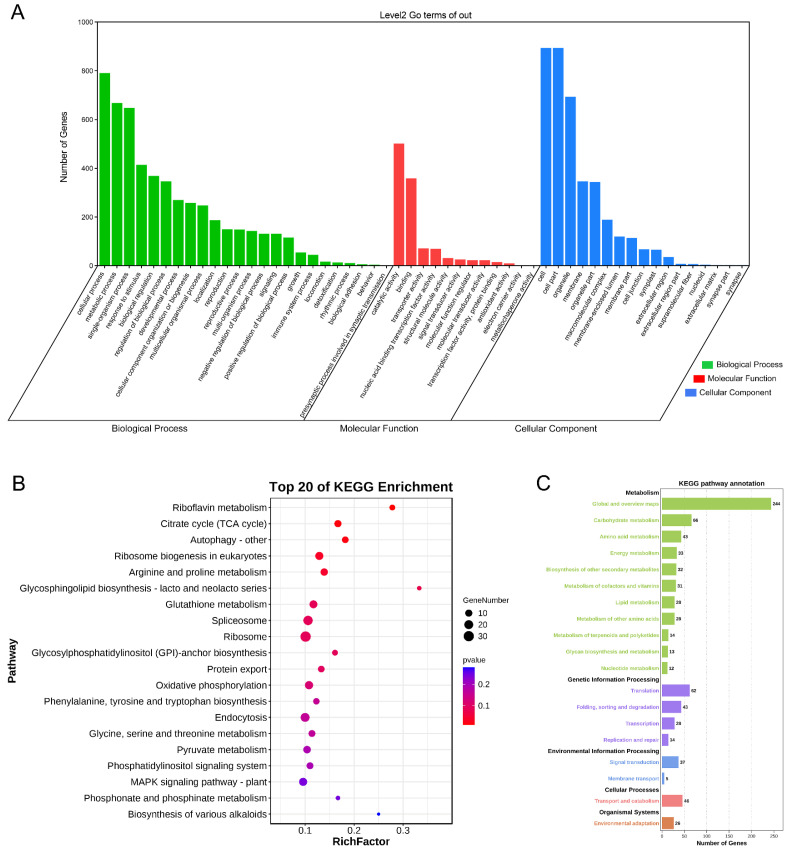
Gene ontology annotation and KEGG enrichment analysis of *BnAP2/ERF* target genes: (**A**) gene ontology (GO) annotation of *BnAP2/ERF* target genes; (**B**) top 20 of KEGG enrichment; (**C**) KEGG pathway annotation.

**Figure 8 ijms-23-15117-f008:**
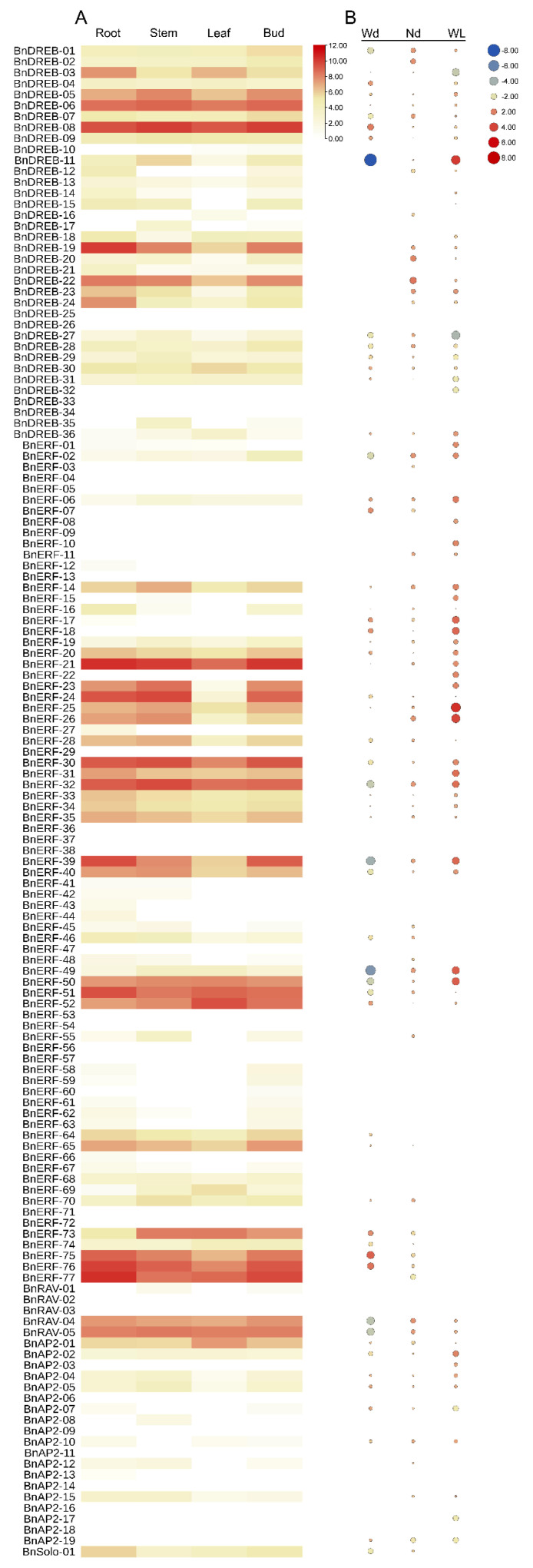
Expression profiles of 138 *BnAP2/ERF* genes based on RNA-seq data: (**A**) The heat map of the expression profiles of *BnAP2/ERF* in four different tissues, which was generated using TBtools software based on log_2_(FPKM + 1) conversion counts of RNA-seq data. Red and white boxes indicate high and low expression levels of *BnAP2/ERFs*, respectively. (**B**) The heat map of the relative expression (converted by log_2_FC values) of *BnAP2/ERF* under water deficit (Wd), nitrogen deficit (Nd), and waterlogging (WL), with up-regulated genes marked in red and down-regulated genes marked in blue.

**Figure 9 ijms-23-15117-f009:**
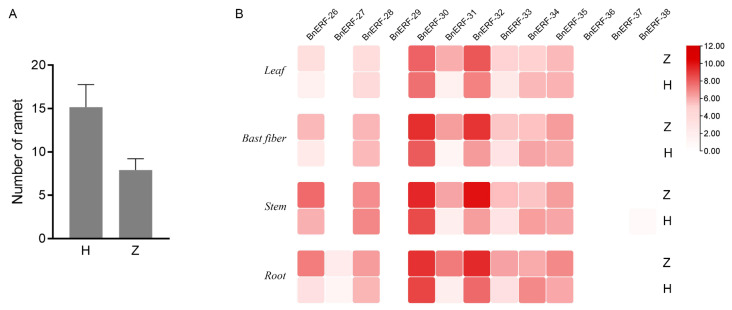
(**A**) Differences in the number of ramets of two ramie varieties. H denotes the ramie variety “He jiang qing ma” with a high number of ramet. Z denotes the ramie variety “Zhongzhu No.1” with a low number of ramet. (**B**) Relative expression profiles of group VIII members of BnAP2/ERF TFs in different tissues of the two ramie varieties.

**Figure 10 ijms-23-15117-f010:**
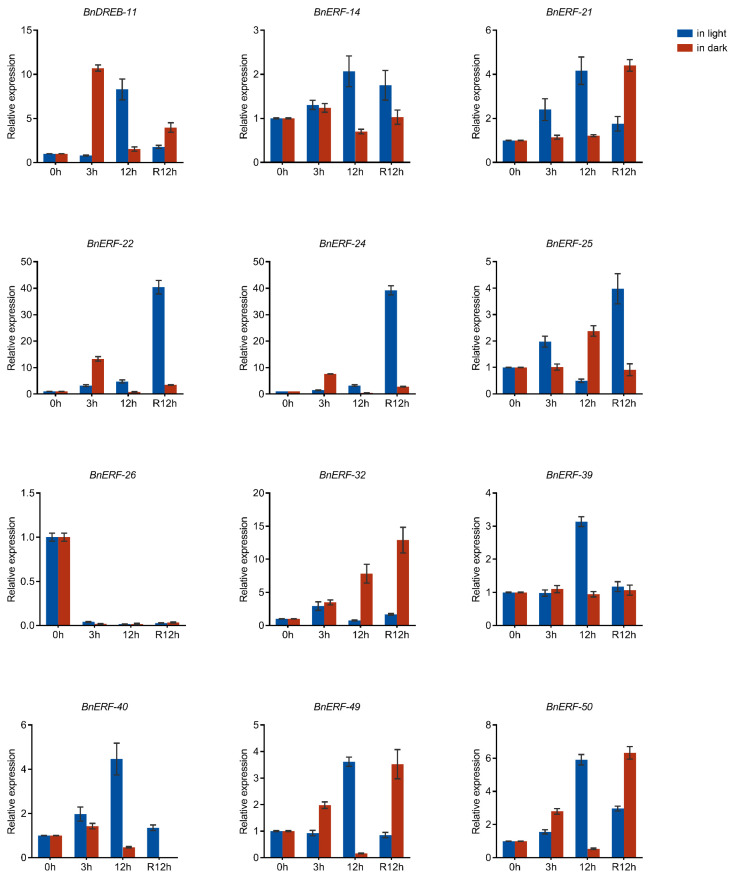
The qPCR results of selected *BnAP2/ERF* genes in ramie leaf samples at 0, 3, and 12 h after waterlogging, and at 12 h (R12h) recovery to normal growth condition. The blue and red bars represent the relative expression level of *BnAP2/ERF* genes of ramie under light and dark conditions, respectively. The error bars represent the standard deviation of three biological duplicates.

**Figure 11 ijms-23-15117-f011:**
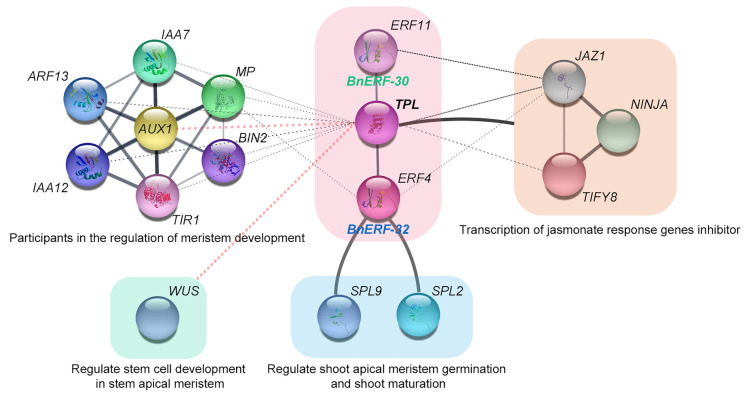
A BnAP2/ERF protein interaction network based on *Arabidopsis* homologs.

**Table 1 ijms-23-15117-t001:** The profile and comparison of AP2/ERF superfamily TF genes between ramie and Arabidopsis.

	Arabidopsis		Ramie	
Classification	Group	No.	Group	No.
AP2 family	Total	18	Total	19
	Double AP2/ERF domain	14	Double AP2/ERF domain	13
	Single AP2/ERF domain	4	Single AP2/ERF domain	6
ERF family	Total	122	Total	113
DREB subfamily	Groups I to IV	57	Groups I to IV	36
ERF subfamily	Groups V to X	58	Groups V to X	75
ERF subfamily	Groups VI-L and Xb-L	7	Groups VI to L	2
Soloist	At4g13040	1	BnSolo-01	1
RAV family		6		5
Total		147	Total	138

## Data Availability

The datasets used and/or analyzed during the current study are available from the corresponding author on reasonable request.

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
