# Peer review of "Genome-Wide Analysis of AP2/ERF Gene Superfamily in Ramie (Boehmeria nivea L.) Revealed Their Synergistic Roles in Regulating Abiotic Stress Resistance and Ramet Development"

_ijms, 2022, doi:10.3390/ijms232315117_

Round 1

Reviewer 1 Report

The manuscript entitled “Genome-wide identification and characteristics of AP2/ERF gene superfamily in ramie (Boehmeria nivea L.) revealed their multifunctional roles in regulating abiotic stress resistance and plant growth”. The manuscript described the identification and functional analysis of AP2/ERF transcription factors (TFs) in ramie, the manuscript is interesting, but I have some suggestions for authors before further processing the manuscript.

1.     The title of the manuscript should be rephrased.

2.     It is unclear how the authors got the transcriptome data; they did RNA-sequencing or got the published data from the database, which need to be explained with references.

3.     What caused the differential expression of BnAP2/ERF in two ramie varieties?

4.     In figure 10, the authors used the blue and red bars; they must indicate which one is for light and dark.

5.     In figure 11, based BnAP2/ERF protein interaction network, the linked genes have strong phenotypes, is the author finding any phenotypical differences in the stress conditions?

6.     For the stress treatments, the author needs to add references; they used the previous literature or optimize the treatment protocols? Need explanation.

Author Response

Response to Reviewer 1 Comments

Dear reviewer,

We would like to thank you for your careful reading, helpful comments, and constructive suggestions, which has significantly improved the presentation of our manuscript.

We have carefully considered all comments from the reviewers and revised our manuscript accordingly. The manuscript has also been double-checked, and the typos and grammar errors we found have been corrected. In the following section, we summarize our responses to each comment from the reviewers. We believe that our responses have well addressed all concerns from the reviewers. We hope our revised manuscript can be accepted for publication.

Q1: The title of the manuscript should be rephrased.

Response: We are grateful for the suggestion and have rephrased the title as “Genome-wide identification and characteristics of AP2/ERF gene superfamily in ramie (Boehmeria nivea L.) revealed their multifunctional roles in regulating abiotic stress resistance and ramet development”.

Q2: It is unclear how the authors got the transcriptome data; they did RNA-sequencing or got the published data from the database, which need to be explained with references.

Response: Thank you for pointing out this issue. Five transcriptomes were used in this manuscript, which are recent RNA-seq data from our lab. Three of these transcriptomes, tissue specificity, waterlogging and two varieties with differential ramet numbers, were unpublished. The transcriptomes of water deficit and nitrogen deficit have been published, but the raw data was not uploaded to the online database. Thus, we just cited the literature in the manuscript. And we have rephrased the method in the manuscript on Line 495 – Line 497.

Q3: What caused the differential expression of BnAP2/ERF in two ramie varieties?

Response: Thank you for your query about the manuscript. For the cause of gene differential expressions in two ramie varieties, it is possible that sequence differences in the promoter regions. The results of our study showed that the BnAP2/ERF promoter regions are rich in cis-acting elements that play critical roles in the transcriptional activity of genes. BnAP2/ERFs in different varieties of ramie may have differences in cis-acting element components in promoter regions, which affect the gene transcriptional levels. Moreover, post-transcriptional regulation is also a cause of gene differential expression.

Q4: In figure 10, the authors used the blue and red bars; they must indicate which one is for light and dark.

Response: Thank you for pointing out this issue. We have added legends to the figure 10. It makes the figure more clear and concise.

Q5: In figure 11, based BnAP2/ERF protein interaction network, the linked genes have strong phenotypes, is the author finding any phenotypical differences in the stress conditions?

Response: Thank you for your query! The query you give us is what we are going to do next. In this study, we only performed a functional prediction of the BnAP2/ERF family genes, especially the two key genes BnERF-30 and BnERF-32, and found that they are coupling regulated in waterlogging stress and ramet development. This result was also demonstrated by the construction of a protein interaction network through homologs in Arabidopsis. However, further experiments are needed to verify whether these interacting proteins have the same behavior in ramie. Our laboratory is constructing a yeast library of ramie, and the next step will be to validate BnERF-30 and BnERF-32 with these interacting proteins and the mechanism of their synergistic regulation in waterlogging tolerance and ramet development. And we are very glad to share the findings in the future.

Q6: For the stress treatments, the author needs to add references; they used the previous literature or optimize the treatment protocols? Need explanation.

Response: Thank you for pointing out this issue. In this study, the waterlogging treatment method was implemented with reference to [1] and [2], and the treatment time was obtained after several experiments and optimization. We have noted and cited specific references on Line 517 – Line 519 and Line 520 - Line 521. Moreover, we offer you our sincerest apologies for the redundancy of water deficit and nitrogen deficit stress treatment methods. Water deficit and nitrogen deficit stress analyses were eventually presented using the transcriptome data of [3]. Therefore, this part of the method is redundant and we considered removing it to avoid confusion. We are sorry for this.

Addition:

  1. We re-phrased the section on Line 118 – Line 125, to make it easier to understand. And also modified the figure 3.
  2. To make the article more logical, we moved this sentence to the first paragraph of the discussion (Line 310 – Line 322)

Reference:

  1. Ren, B.; Hu, J.; Liu, P.; Zhao, B. and Zhang, J. Responses of nitrogen efficiency and antioxidant system of summer maize to waterlogging stress under different tillage. PeerJ. 2021, 9: e11834. 10.7717/peerj.11834
  2. Luo, F.L.; Nagel, K.A.; Zeng, B.; Schurr, U. and Matsubara, S. Photosynthetic acclimation is important for post-submergence recovery of photosynthesis and growth in two riparian species. Ann Bot. 2009, 104(7): 1435-1444. 10.1093/aob/mcp257
  3. Chen, J.; Gao, G.; Chen, P.; Chen, K.; Wang, X.; Bai, L.; Yu, C. and Zhu, A. Integrative Transcriptome and Proteome Analysis Identifies Major Molecular Regulation Pathways Involved in Ramie (Boehmeria nivea (L.) Gaudich) under Nitrogen and Water Co-Limitation. Plants (Basel). 2020, 9(10). 10.3390/plants9101267

Thank you for your precious comments and advice. Those comments are all valuable and very helpful for revising and improving our paper. We have revised the manuscript accordingly, and our point-by-point responses are presented above. We are looking forward to hearing from you soon.

Sincerely yours

Reviewer 2 Report

The authors made a deep characterization of the AP2/ERF gene family on B. nivea, a gene family of transcription factors that is fundamental in many aspects of plant development and growth. The manuscript is well articulated, it follows a logic organization and the problem is well addressed. The results section is perfectly logic; the figures are complete and well designed. I have some problem due to the size of some of the figures; however, I believe that is due to the manuscript format that is available for reviewers and not a real problem for the final version. The description of some figures is an aspect that I believe it could be improved, I have some suggestion that I enlisted at the end of these comments.

The only section that I did not found adequate enough is the discussion, I found it incomplete since many of the aspects mentioned in the results section are not even mentioned. Discussion for the gene duplication outcomes, the cis-elements detected, the KEGG enrichment analysis is missing. The biological relevance of these outcomes must be highlighted by the authors to justify its inclusion on the main part of the paper.

Here is the list of minor changes that I suggest to improve the quality of the article:

Line 22 – Correct the word “duplication” for: duplication.

Line 49 – Line 51 – It’s necessary to re-phrase this paragraph, is not clear enough.

Line 59 – Change “involved in plant tiller” for: involved in plant patterns of tillering.

Line 80 till Line 99 – In this paragraph, the authors use the word gene instead of the word protein. Since the phylogenetic analysis was performed with the aminoacid sequences, the authors should use the word protein.

Line 84 – Abbreviations are used here for the first time, but the complete form is explained much later in the article (Line 323). Similar situations can be found throughout the manuscript, please modify them all.

Line 115 – Change “which demonstrated their important function” for: which suggest their important function.

Line 116-117 – I do not understand why is noticeably that only BnRAV-04 is the only gene with introns? Can you explain it further or re-phrase it?

Line 123-128 – Please re-phrase this whole paragraph since it not understandable in its actual form.

Line 182-183 – I believe this information is misleading since it can be interpreted as the trans-elements are not part of such processes. Please re-write it.

Line 185 – Change “phytohormoneresponsive” for:  phytohormone responsive.

Line 201 – “the size of the circles”. Should be: the size of the squares.

Line 227 – Please re-phrase this paragraph. Change “Thirty nine genes were not expressed in all tissues” for: Thirty nine genes were not expressed in any analysed tissue.

Line 240 – Section 2.6.2. In my opinion, the genes which expression is detected only in situations of abiotic stress are among the most interesting in this family. However, the authors do not highlight them enough and some of them are not even described or barely mentioned (BnDREB32 & 34, BnERF17 &18). Include them also in the discussion section, does some of the orthologs from Arabidopsis show the same behaviour?

Line 266-276. Please improve the writing of this section since it is very difficult to read on its actual form.

Line 286-289. A more detailed explanation of how those 12 genes were selected for qPCR analysis is required. The selection process is not clear enough.

Line 294 – The authors mentioned that “BnDREB-11, BnERF-39, and BnERF-49 decreased before increasing”, but I cannot detect that pattern for these genes, specifically the expression decrease is not detectable.

Line 302-304 – “BnERF-26 showed strong induction under stress as well as in the recovery state after stress, and it is speculated that it may be important in the growth and development of plants under stress.”. The expression of this gene is almost no detectable during abiotic stress or recovery. And this information is also contradictory with the information presented on line 309: “The expression of BnERF-26 was strongly inhibited under stress as well as during the recovery stage.”.

Line 329 -  Change “(E19/d19)” for: (E19/D19).

Line 336 – Change “Therefore, we believe that AP2/ERF” for: Therefore, we hypothesize that AP2/ERF.

Line 388-392 – This paragraph is not clear enough, please re-phrase it.

Line 393 – A better explanation of why BnERF-30 and BnERF32 were chosen for the protein interaction analysis is needed. The given explanation is not clear enough cause many of the other genes presented similar expression pattern.

Author Response

Response to Reviewer Comments

Dear reviewer,

We would like to thank you for your careful reading, helpful comments, and constructive suggestions, which has significantly improved the presentation of our manuscript.

We have carefully considered all comments from the reviewers and revised our manuscript accordingly. The manuscript has also been double-checked, and the typos and grammar errors we found have been corrected. In the following section, we summarize our responses to each comment from the reviewers. We believe that our responses have well addressed all concerns from the reviewers. We hope our revised manuscript can be accepted for publication.

Q1: Line 22 – Correct the word “duplication” for: duplication.

Response: We apologize for the spelling problems in the original manuscript. We have corrected the word at Line 22 in the manuscript.

Q2: Line 49 – Line 51 – It’s necessary to re-phrase this paragraph, is not clear enough.

Response: We are grateful for the suggestion. In order to make this passage more clearly, we have re-phrased it on Line 48 - Line 50.

Q3: Line 59 – Change “involved in plant tiller” for: involved in plant patterns of tillering.

Response: We agree with the comment and re-wrote the sentence in the revised manuscript as the following: involved in plant patterns of tillering (Line 57 – Line 58).

Q4: Line 80 till Line 99 – In this paragraph, the authors use the word gene instead of the word protein. Since the phylogenetic analysis was performed with the amino acid sequences, the authors should use the word protein.

Response: Thank you for pointing out this problem in our manuscript. We wrote the manuscript with an emphasis on transcription factors and genes, thus neglecting this point. Thank you again for pointing this out, and we have corrected this section on Line 78 - Line 94.

Q5: Line 84 – Abbreviations are used here for the first time, but the complete form is explained much later in the article (Line 323). Similar situations can be found throughout the manuscript, please modify them all.

Response: Thank you for pointing this issue, this is our negligence. We have put the complete form of the reference words at the first time when they were mentioned (Line 80 - Line 83), and another similar issue “differential expressed genes (DEGs)” was corrected on Line 230 and Line 233.

Q6: Line 115 – Change “which demonstrated their important function” for: which suggest their important function.

Response: We are grateful for the suggestion, and we have changed the word on Line 112.

Q7: Line 116-117 – I do not understand why is noticeably that only BnRAV-04 is the only gene with introns? Can you explain it further or re-phrase it?

Response: Thank you for pointing out this issue. We did neglect to explain this point. By learning the AP2/ERF family in other plants, we found that there are four RAV subfamily members in ramie, three of which contain introns, and only one gene, BnRAV-04, does not contain introns, which is different from most RAV subfamily members in other plants without introns. This phenomenon may have occurred because ramie evolved to adapt to environmental stresses resulting in the insertion of more introns to maintain the stability of transcription of these genes. In response to this issue, we explained this phenomenon in the discussion section (Line 331 - Line 338).

Q8: Line 123-128 – Please re-phrase this whole paragraph since it not understandable in its actual form.

Response: Thank you for pointing out this issue. After careful consideration of this section, we finally decided that this section is an explanation and discussion of the exclusive motif, MCCGGAI(I/L) motif and EAR motif, which contained in group VII and group VIII, respectively. This section is more suitable for discussion, so we have moved them to the discussion section (Line 353 – Line 363) and added to the content.

Q9: Line 182-183 – I believe this information is misleading since it can be interpreted as the trans-elements are not part of such processes. Please re-write it.

Response: Thank you for pointing out this issue. After careful consideration, it is true that this sentence is somewhat misleading and redundant. Therefore, we deleted this sentence (Line 166).

Q10: Line 185 – Change “phytohormone responsive” for: phytohormone responsive.

Response: We apologize for the formatting error again, and we have corrected it on Line 168.

Q11: Line 201 – “the size of the circles”. Should be: the size of the squares.

Response: Thank you for pointing out this issue. This error occurred because we remade a visualization analysis for this section but neglected to modify the diagram notes. We have modified it to "color concentration of the squares" according to the meaning shown in the figure 6 (Line 182 – Line 183).

Q12: Line 227 – Please re-phrase this paragraph. Change “Thirty-nine genes were not expressed in all tissues” for: Thirty-nine genes were not expressed in any analyzed tissue.

Response: We agree with the comment and re-phrase the sentence in the revised manuscript as the following: Thirty-nine genes were not expressed in any analyzed tissue (Line 209).

Q13: Line 240 – Section 2.6.2. In my opinion, the genes which expression is detected only in situations of abiotic stress are among the most interesting in this family. However, the authors do not highlight them enough and some of them are not even described or barely mentioned (BnDREB32 & 34, BnERF17 & 18). Include them also in the discussion section, does some of the orthologs from Arabidopsis show the same behavior?

Response: Thank you for pointing out this problem in manuscript. To ensure the correctness and completeness of the analysis in this section, we disposed and reanalyzed the transcriptome data. Then we re-phrase and add the content in the revised manuscript (Line 239 – Line 250, Line 385 – Line 392).

Q14: Line 266-276. Please improve the writing of this section since it is very difficult to read on its actual form.

Response: Thank you for pointing out this issue. We read this passage over and over again and found it truly incomprehensible. So, we reanalyzed and re-phrased this section (Line 253 – Line 266) to make it easier to read and understand, and re-visualize the figure 9.

Q15: Line 286-289. A more detailed explanation of how those 12 genes were selected for qPCR analysis is required. The selection process is not clear enough.

Response: Thank you for your advice and suggestion. We have re-phrased and added to this section, and the general content and reasons are as follows: we first selected members of the VIII group (BnERF-21 to BnERF25) in ramie as candidate genes based on the reported members of group VIII involved in waterlogging stress in Arabidopsis [1]. Since the two genes BnERF-22 and BnERF-23 differ by only 13 base deletions, BnERF-23 could not be designed with specific primers, so BnERF-23 was removed. In addition, we selected genes with higher expression (FPKM > 20) as well as greater variability from the transcriptome of waterlogging stress as candidate genes. The 12 genes in the manuscript were finally selected for qPCR analysis. The concrete content re-phrased is on Line 272 – Line 278.

Q16: Line 294 – The authors mentioned that “BnDREB-11, BnERF-39, and BnERF-49 decreased before increasing”, but I cannot detect that pattern for these genes, specifically the expression decrease is not detectable.

Response: Thank you for pointing out this problem. We are sorry to have misinterpreted the changes in the expression of these genes. We found by re-analysis of the data that their expression at 3 h did not show a significant trend from the control. We have corrected and re-phrase the sentences on Line 283 and Line 284.

Q17: Line 302-304 – “BnERF-26 showed strong induction under stress as well as in the recovery state after stress, and it is speculated that it may be important in the growth and development of plants under stress.”. The expression of this gene is almost no detectable during abiotic stress or recovery. And this information is also contradictory with the information presented on line 309: “The expression of BnERF-26 was strongly inhibited under stress as well as during the recovery stage.”.

Response: Thank you for pointing out this issue in manuscript. Here we used the inappropriate word “induction” and made it misunderstanding. We have corrected and re-phrased the sentences in the manuscript (Line 292 – Line 293 and Line 298 – Line 299).

Q18: Line 329 - Change “(E19/d19)” for: (E19/D19).

Response: We apologize for the spelling problems in the manuscript again. We would like to express our sincere gratitude for your careful review of the manuscript. We have modified the letter (Line 343) and carefully reviewed the entire manuscript for this same mistake.

Q19: Line 336 – Change “Therefore, we believe that AP2/ERF” for: Therefore, we hypothesize that AP2/ERF.

Response: Thank you for your suggestion, and we have changed the word to “hypothesize” on Line 361.

Q20: Line 388-392 – This paragraph is not clear enough, please re-phrase it.

Response: Thank you for your suggestion, and we have re-phrased this section and removed some redundancies, “In particular, when plants are under anoxic conditions such as waterlogging stress, the root system will maintain vital signs by increasing glycolytic flux. This series of mechanisms comes at the cost of a severe decrease in fiber production. Moreover, the ramet number is a vital component of ramie yield and the ramet ability is decisive for high yield of ramie”, to make it logical and easy to understand (Line 378 – Line 385).

Q21: Line 393 – A better explanation of why BnERF-30 and BnERF32 were chosen for the protein interaction analysis is needed. The given explanation is not clear enough cause many of the other genes presented similar expression pattern.

Response: Thank you for your advice and suggestion. We have re-phrased this section and added some necessary contents to explain why we chose these two genes, BnERF-30 and BnERF-32, for the protein interaction analysis (Line 415 – Line 425).

Addition:

  1. We re-phrased the section on Line 118 – Line 120, to make it easier to understand. And also modified the figure 3.
  2. To make the article more logical, we moved this sentence to the first paragraph of the discussion (Line 310 – Line 322)

Reference:

  1. Zhou, Y.; Tan, W.J.; Xie, L.J.; Qi, H.; Yang, Y.C.; Huang, L.P.; Lai, Y.X.; Tan, Y.F.; Zhou, D.M.; Yu, L.J.; Chen, Q.F.; Chye, M.L. and Xiao, S. Polyunsaturated linolenoyl-CoA modulates ERF-VII-mediated hypoxia signaling in Arabidopsis. J Integr Plant Biol. 2020, 62(3): 330-348. 10.1111/jipb.12875

Thank you for your precious comments and advice. Those comments are all valuable and very helpful for revising and improving our paper. We have revised the manuscript accordingly,and our point-by-point responses are presented above. We are looking forward to hearing from you soon.

Sincerely yours

Round 2

Reviewer 1 Report

The authors have improved the manuscript quality, but the title still needs improvement.

Author Response

Response to Reviewer 1 Comments

Dear reviewer,

We are very grateful to your comments for the revised manuscript. According to your advice, we have tried our best to improve and made some changes in the manuscript. About the English writing of the manuscript, we have asked native English speakers to check it. All corrections in the manuscript were marked in red font.

Comment 1: The authors have improved the manuscript quality, but the title still needs improvement.

Response: Thank you a lot for your comments. After discussion by all our authors, we decided to change the title to “Genome-wide analysis of AP2/ERF gene superfamily in ramie (Boehmeria nivea L.) revealed their synergistic roles in regulating abiotic stress resistance and ramet development”

In addition: We have re-phrase the conclusions on Line 535 – Line 549.

Thank you for your precious comments and advice, which rounded out our manuscript. We have revised the manuscript accordingly and look forward to your reply.

Sincerely yours
